# Valley-Dependent Electronic Properties of Metal Monochalcogenides GaX and Janus Ga_2_XY (X, Y = S, Se, and Te)

**DOI:** 10.3390/nano14151295

**Published:** 2024-07-31

**Authors:** Junghwan Kim, Yunjae Kim, Dongchul Sung, Suklyun Hong

**Affiliations:** Department of Physics, Graphene Research Institute, Quantum Information Science and Technology Center, Sejong University, Seoul 05006, Republic of Korea

**Keywords:** valleytronics, metal monochalcogenides (MMC), Janus MMC, Berry curvature, DFT calculations

## Abstract

Two-dimensional (2D) materials have shown outstanding potential for new devices based on their interesting electrical properties beyond conventional 3D materials. In recent years, new concepts such as the valley degree of freedom have been studied to develop valleytronics in hexagonal lattice 2D materials. We investigated the valley degree of freedom of GaX and Janus GaXY (X, Y = S, Se, Te). By considering the spin–orbit coupling (SOC) effect in the band structure calculations, we identified the Rashba-type spin splitting in band structures of Janus Ga_2_SSe and Ga_2_STe. Further, we confirmed that the Zeeman-type spin splitting at the K and K’ valleys of GaX and Janus Ga_2_XY show opposite spin contributions. We also calculated the Berry curvatures of GaX and Janus GaXY. In this study, we find that GaX and Janus Ga_2_XY have a similar magnitude of Berry curvatures, while having opposite signs at the K and K’ points. In particular, GaTe and Ga_2_SeTe have relatively larger Berry curvatures of about 3.98 Å^2^ and 3.41 Å^2^, respectively, than other GaX and Janus Ga_2_XY.

## 1. Introduction

Since electronic devices were first developed, one has tried to control the electrons in materials more precisely. By handling the charge and spin of an electron in three-dimensional (3D) materials, we have made various types of electronic devices such as transistors, diodes, magnetic memory devices, and so on. In 2004, it was discovered that graphene shows excellent atomic, mechanical and electronic properties [1,2,3,4,5]. Since the discovery of graphene, various 2D materials have received much attention from the scientific community because of their excellent atomic and electronic properties. Particularly, many researchers have mainly focused on making electronic devices by using 2D materials to replace or complement conventional 3D materials, and thus have shown their outstanding potential for electronic devices based on their interesting electrical properties beyond conventional 3D materials [6,7,8,9,10,11,12,13,14,15,16,17]. In addition, as research on various 2D materials progressed, studies on Janus 2D materials, which comprise sandwich configurations with different atoms, were also conducted. The absence of inversion symmetry in a Janus 2D monolayer materials leads to high piezoelectric properties. Due to these characteristics, research is being conducted on the piezoelectric properties of these materials, as well as on their thermoelectric properties and phototactic behaviors [18,19,20,21,22,23,24,25].

Recently, a new concept of ‘valleytronics’ based on the valley degree of freedom has been suggested in relation to hexagonal lattice 2D materials. The valley degree of freedom, which is a new method to control electrons, means that there are maximum or minimum points of band structures having the same energy at different momentum values. Interestingly, the valley degree of freedom in 2D materials is free of intervalley scattering since the positions of valleys in momentum space are far separated from each other. At valleys, Zeeman-type spin splitting by spin–orbit coupling (SOC) gives the possibility of significant spin-valley coupling [26,27]. In addition to Zeeman-type spin splitting, Rashba-type spin splitting driven from the potential gradient or electric field induced by broken mirror inversion symmetry has been studied for valleytronics and spintronics [26,27]. Recently, because valleys naturally exist in band structures of various materials, there have been many experimental reports concerning the control of electrons through the valley degree of freedom [28,29,30,31,32]. Optical valley pumping is the first way to do so. In transition metal dichalcogenides (TMDs) with the formula MX_2_ (M = TM, X = chalcogens), two valleys at the K and K′ points have the same energy values with contributions by opposite spins due to the time reversal symmetry, and Zeeman-type spin splitting at the K and K′ points is large due to the orbital magnetic moment resulting from the orbital motion of electrons [33,34,35]. Thus, different spin contributes to each valley at the K and K′ points, and one can experimentally excite the electrons through a specific valley using polarized light [36,37]. The second is the valley Hall effect. Interestingly, as the Hall effect is controlled by a Lorentz force proportional to the magnetic field on a moving electron, the formula describing the valley Hall effect is related to the cross product of the electric field and Berry curvature that acts like a magnetic field on moving electrons. Note that since the Berry curvature is defined by the crystal structure of a given material, an external magnetic field does not have to be applied to the material to obtain the valley Hall effect [38,39,40,41]. Based on various methods of controlling electrons through valleys including the two methods mentioned above, 2D materials like TMDs and Janus TMDs (MXY, X, Y = chalcogen) have been proposed as candidates for valleytronics [42,43,44,45]. The relation between valleytronics and TMD materials has been thoroughly explored in various studies [46,47]. However, until now, most studies of valleytronics of 2D materials have been limited to TMDs, while there are few reports of other 2D materials like metal monochalcogenides (MMCs).

In this study, among various 2D materials, we focus on MMCs and Janus MMCs showing good physical and electrical properties for device applications. Specifically, we consider GaX (or Ga_2_X_2_) and Janus Ga_2_XY (X, Y = S, Se, Te), the electronic structures of which are investigated for future valleytronic device applications using density functional theory (DFT) calculations. The details of the electronic structure are studied using band structure and Berry curvature calculations.

## 2. Calculation Methods

For investigations of GaX and Janus Ga_2_XY, we perform DFT calculations within the generalized gradient approximation (GGA) for the exchange-correlation (xc) functionals [48,49], implemented in the Vienna ab initio simulation package (VASP). The kinetic energy cut-off is set to 400 eV, and the projector-augmented wave potentials (PAW) are used to treat the ion-electron interaction [50,51,52,53]. For the van der Waals (vdW) corrections, we use Grimme’s DFT-D3 method based on a semiempirical GGA-type theory [54]. To remove artificial interaction between the adjacent slabs, we use a vacuum region of about 20 Å. The SOC effect is considered in the electronic structure calculations. In the DFT calculations, we consider (1 × 1) hexagonal unit cells of monolayer GaX and Janus Ga2XY. Furthermore, we use the (12 × 12 × 1) grid in the gamma-centered scheme for the Brillouin zone integration. The Hellmann–Feynman forces and energy convergence criteria are set to 0.01 eV/Å and 10–5 eV for ionic relaxation, respectively. Moreover, to study the thermal stability of GaX and Janus Ga_2_XY, we perform an ab initio molecular dynamics (AIMD) simulation. The NVT ensemble is considered, and the simulation temperature is set to 300 K. The (6 × 6) hexagonal unit cell for AIMD simulation is used, and the total simulation time is set to 5000 fs, while each time step is 1 fs.

The Berry curvature is a fundamental physical quantity that plays a pivotal role in the study of topological materials. The Berry curvature $\Omega_{ij}$ is defined as $\Omega_{ij}=\partial_{i}A_{j}-\partial_{j}A_{i}$, where $\partial_{i}$ and $\partial_{j}$ are partial derivative operators with respect to parameters in the parameter space, and $A_{i}$ and $A_{j}$ are the components of the Berry connection. The Berry connection $A_{i}$ is defined as $A_{i}=i\langle u\left(k\right)|\partial_{i}u\left(k\right)\rangle$, where $|\left.u\left(k\right)\right\rangle$ represents the periodic part of the Bloch state, which encapsulates the geometric properties of the electronic state, and k is the wavevector in the Brillouin Zone. The Berry curvature can be expressed using Bloch states as follows [55,56,57]:$$\Omega_{ij}=i\left(\langle \partial_{i}u\left(k\right)|\partial_{j}u\left(k\right)\rangle -\langle \partial_{j}u\left(k\right)|\partial_{i}u\left(k\right)\rangle \right)$$

In our calculations, we use VASPBERRY software (ver. 1.0, Jülich, Germany) [58] to investigate the Berry curvature for the Janus GaX and Ga_2_XY materials.

## 3. Results and Discussion

The structures of monolayer GaX and Janus Ga_2_XY have two Ga atoms at the center and two chalcogen atoms located outside of the Ga atoms in their hexagonal unit cell. Figure 1 shows the optimized structure of Janus Ga_2_XY, which can be obtained by replacing the X atom of GaX (=Ga_2_X_2_) with a Y atom. The structural parameters of monolayer GaX and Janus Ga_2_XY, such as lattice constants, bond lengths and angles, are given in Table 1. The results show that the lattice parameters (*a*) of Janus Ga_2_XY are close to the average values of the lattice parameters of GaX and GaY. The bond length between Ga atoms (*d*_Ga-Ga_) does not change depending on the type of chalcogen atom. The bond lengths (*d*_Ga-X,_ *d*_Ga-Y_) and angles (*θ*_1_ and *θ*_2_) increase when the atomic number of chalcogen atoms is increased. Clearly, the structural mirror inversion symmetry of Janus Ga_2_XY is broken. Next, we calculate the plane-averaged local potentials of monolayer GaX and Janus Ga_2_XY. As shown in Figure 2, the difference in local potential (△φ) between the upper and lower sides of GaX is zero for every GaX. However, the △φ values of Ga_2_XY are nonzero, which means that the charge transfer occurs between the X and Y atoms and an internal dipole moment in the out-of-plane direction is induced in Ga_2_XY. To examine the charge transfer analysis, the Bader charge analysis method was employed. In the Badger charge analysis of Ga_2_XY, a charge redistribution from Ga atoms to X and Y atoms was observed, indicating that the △φ can be attributed to the charge transfer mechanism. Because the △φ values are obviously proportional to the dipole moment [59] and represent the difference in work function and also the potential gradient of GaX and Ga_2_XY, we confirm that Ga_2_STe has the largest internal dipole moment among the Ga_2_XY, as shown in Table 1. In particular, internal dipole moment for the Janus Ga_2_XY materials occurs from the higher number of chalcogen atoms to the lower number of chalcogen atoms.

To study the thermal stability of monolayer GaX and Janus Ga_2_XY, we perform ab-initio molecular dynamics (AIMD) calculations. The temperature is set at room temperature (RT, 300 K), while the time step and the total simulation time are 1 fs and 5000 fs, respectively. After relaxation at RT, we plot the total energies as a function of time in Figure 3. Only a small range of fluctuation in total energy is shown, and there is no structural destruction in GaX and Ga_2_XY after 5000 fs, which implies that GaX and Ga_2_XY have good thermal stability at RT. The stability of Ga_2_XY and other Janus materials was demonstrated by previous results and supported by phonon calculations [60,61,62]. Recently, by using the chemical vapor deposition (CVD) method, the successful synthesis of 2H Janus MoSSe was reported [63,64]. In this regard, we expect that Janus Ga_2_XY with a 2H structure like Janus MoSSe can be synthesized by the same experimental method used for making Janus MoSSe.

Next, we investigate the electronic structure of GaX and Janus Ga_2_XY by using band structure calculations. For the calculations below, the hexagonal Brillouin zone for GaX and Janus Ga_2_XY are used, as shown in Figure 4a, while schematic diagrams for Rashba-type spin splitting, Zeeman-type spin splitting, and the directional change in valley spins depending on the Berry curvature are given in Figure 4b–d, respectively. To study the valley-contrasting effect at the K and K′ points, we choose a symmetry path including K and K′. Figure 5 presents band structures of GaX and Janus Ga_2_XY along the symmetric path, obtained by considering SOC interaction. The up- and down-spin contributions determined by the size of circle are represented by red and blue colors, respectively. The valence band maximum (VBM) points of GaX and Ga_2_SSe are located between the K and Γ points, while Ga_2_STe and Ga_2_SeTe have VBM points at the Γ point. In the conduction band, all GaX and Ga_2_XY, except for GaTe, have the conduction band minimum (CBM) at the Γ point. These VBM and CBM points align closely with those depicted in both experimental and theoretical results [65,66]. Thus, GaX and Janus Ga_2_SSe have indirect band gaps with VBM between the K and Γ points and CBM at the Γ point, but the Janus GaSTe and GaSeTe have direct band gaps with both VBM and CBM located at the Γ point. In addition, the order of band gap size is found to be GaS (2.46 eV) > Ga_2_SSe (2.16 eV) > GaSe (1.89 eV) > GaTe (1.50 eV) > Ga_2_SeTe (0.98 eV) > Ga_2_STe (0.64 eV). Indeed, GaX and Janus Ga_2_XY show a wide range of band gaps.

To understand the SOC effect on band structure calculations, we study Zeeman-type and Rashba-type spin splitting in GaX and Janus Ga_2_XY induced by the presence of the intrinsic out-of-plane electric field or dipole moment due to inversion symmetry breaking. Basically, a strong potential gradient causes the SOC effect, and inversion symmetry breaking of the Janus structure increases the out-of-plane electric field. For example, it was reported that Zeeman-type spin splitting and Rashba-type spin splitting are identified at Janus MoSSe by out-of-plane intrinsic electric fields. Based on these results, MoSSe suggests a great possibility for applications in 2D spintronics and valleytronics [67]. In Janus Ga_2_SSe and Ga_2_STe, the Rashba-type spin splitting indicating band crossing or splitting at the Γ point is found, as shown in Figure 5d,e, but there seems to be no Rashba-type spin splitting in GaX and Ga_2_SeTe. In Janus Ga_2_SSe and Ga_2_STe, the band splitting by Rashba-type spin splitting does not occur at the VBM points. Instead, such band splitting occurs at the bands just below the VBM in both materials. For device applications utilizing the band structure changes induced by the Janus MMC materials, band splitting would be most effective if it occurred at the VBM. While neither material is ideal in this respect, Ga_2_SSe is relatively better for practical use because the energy difference E_R_ is larger in Ga_2_SSe than in Ga_2_STe. In addition, to estimate the amount of Rashba-type spin splitting, we calculate the Rashba coefficient (***α***_R_ = 2*E*_R_/*K*_R_), where *E*_R_ is the Rashba energy defined by the VBM energy measured from the energy at Γ point, and K_R_ is the momentum offset defined by the momentum distance from the Γ point to VBM point, as shown in Figure 4b. The Rashba coefficients of Janus Ga_2_SSe and Ga_2_STe are 0.678 meV Å and 1.72 meV Å, respectively. In terms of reciprocal space, Janus Ga_2_SSe shows the Rashba effect on Γ point, while the VBM point is not located at K or K′ point, which means that the electrons lying at the Γ point are hard to control, and thus, utilization of the Rashba effect at the Γ point is difficult. In contrast, the VBM points of Janus Ga_2_STe and Ga_2_SeTe are located at Γ points.

In addition to the Rashba-type spin splitting, we obtain the Zeeman-type spin splitting in GaX and Janus Ga_2_XY. It is confirmed that the K and K′ valleys have opposite spin contributions. The valley polarization is one of the useful ways to exploit the valley degree of freedom. Recently, various experimental methods such as optical pumping have been reported and enable one to control spin selectively under the valley optical selection rule [36,37]. As shown in Table 1, the order of magnitude for the Zeeman-type spin splitting is GaS (202 meV) > GaTe (180 meV) > Ga_2_SSe (97 meV) > GaSe (65 meV) = Ga_2_SeTe (65 meV) > Ga_2_STe (43 meV). However, the selectively excited states of K and K′ formed by optical valley pumping generate nonequilibrium distributions of carriers, which is related to carrier lifetime due to carrier recombination. Based on recent studies, GaX and Ga_2_XY monolayers exhibit valley properties comparable to those of TMDs. For instance, the valley-spin filtering mechanism in TMDs is attributed to metal-induced gap states (MIGS). This research contributes to a fundamental understanding and provides design guidelines for efficient valley-spin filter devices based on hexagonal monolayers with broken inversion symmetry [68]. Additionally, the potential of valley-dependent Lorentz forces in strained TMDs to induce valley separation is discussed [69]. This mechanism, coupled with the Zeeman-type spin splitting observed in our GaX and Ga_2_XY systems, suggests potential applications in valleytronic devices where valley-dependent filtering could be utilized.

The few-layer structure of InSe demonstrates a strong Dresselhaus-type SOC induced by its inversion-asymmetric gamma phase [70]. These features are pivotal for the practical application of SOC effects in Janus materials or alloys. The narrower band gap of few-layer crystals may facilitate doping through gating methods for device integration, while the interlayer hopping asymmetry could lead to significant SOC effects, even in sublayer mixed alloys. For these reasons, we investigate the Dresselhaus-type SOC effects on Janus Ga_2_XY materials. We calculate spin-texture to determine whether it exhibits Dresselhaus-type or Rashba-type spin splitting. As depicted in Figure 6, spin splitting occurs in Janus Ga_2_XY, confirming it to be of Rashba-type. To better illustrate spin splitting, spin textures are plotted from the Fermi level at various positions. The split bands display opposite helical spin textures in Rashba-type materials, whereas Dresselhaus spin splitting exhibits parallel spin polarization along k_x_ and k_y_ in the band dispersion curves. Therefore, there is no need to consider Dresselhaus-type spin splitting in our calculations.

For this reason, the Berry curvature Ω(k) has been actively investigated to expand carrier lifetime since a larger Berry curvature implies more difficult recombination and a longer lifetime. As shown in Figure 4d, the Berry curvature causes carriers to have transverse velocity, which is described by $\dot{r}=\frac{1}{\hbar }\frac{\partial E(k)}{\partial k}-\dot{r}\times \Omega (k)$ using the semiclassical picture. Interestingly, since the Berry curvature has an opposite sign to the out-of-plane direction at K and K′, the transverse velocity reduces the recombination of carriers. Figure 7 shows the Berry curvature Ω(k) of GaX and Janus Ga_2_XY, where the positive and negative signs of the Berry curvature are drawn in red and blue, respectively. The Berry curvature values indicate high symmetry, i.e., 3-fold symmetry, from the Γ point. It is confirmed that GaX and Janus Ga_2_XY have the same magnitude but opposite signs to the Berry curvature at K and K′, as shown in the line profile from A to B points in Figure 6. Therefore, excited carriers such as up and down spins have different transverse directions under the electric field. In detail, a comparison of line profiles of the Berry curvature shows that GaTe (3.98 Å^2^) and Ga_2_SeTe (3.41 Å^2^) have bigger Berry curvature values at K and K′ than other GaX and Janus Ga_2_XY.

## 4. Conclusions

In this study, we have investigated the atomic and electronic properties of GaX and Janus Ga_2_XY using first-principles calculations. We found that GaX and Janus Ga_2_XY show structural stability at room temperature using AIMD calculations. Inversion symmetry breaking in Janus Ga_2_XY leads to internal dipole moments, pointing in the out-of-plane direction in the Janus structure. Next, we studied the band structure of GaX and Janus Ga_2_XY where their band gaps range widely from 0.64 eV to 2.64 eV. In addition, Zeeman-type spin splitting and Rashba-type spin splitting were studied by considering the SOC effect on the band structure. In Janus Ga_2_SSe and Ga_2_STe, Rashba-type spin splitting was observed, indicating a band crossing or splitting, while in GaX and Ga_2_SeTe, there is no Rashba-type spin splitting. On the other hand, in GaX and Janus Ga_2_XY, we confirmed opposite spin contributions to the Zeeman-type spin splitting at K and K′. The largest value of the Zeeman-type spin splitting is 202 meV (GaS) and its smallest value is 43 meV (Ga_2_STe). We calculated the Berry curvature, finding that GaX and Janus Ga_2_XY have the same magnitude but opposite sign of Berry curvature at K and K′. Especially GaTe and Ga_2_SeTe show larger Berry curvatures (3.98 Å^2^ and 3.41 Å^2^, respectively) than other GaX and Janus Ga_2_XY. Considering the effect of strain, it is conjectured that the distortion of the Berry curvature occurs due to the distortion observed in the K-space [42]. Our study provides a deep understanding of GaX and Janus Ga_2_XY for use in valleytronics and spintronics applications.

## Figures and Tables

**Figure 1 nanomaterials-14-01295-f001:**
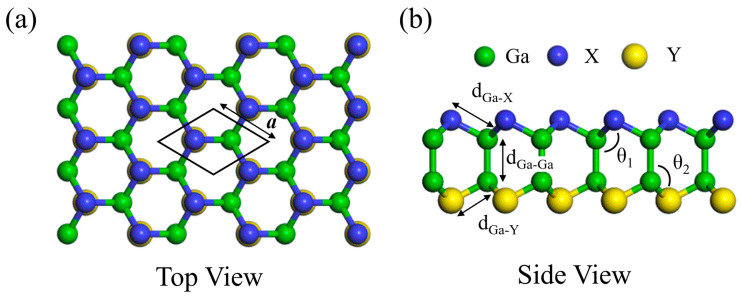
(**a**) The top view and (**b**) side view of monolayer Janus Ga_2_XY. The green, yellow, and blue balls indicate the Ga atoms and chalcogen X and Y atoms, respectively.

**Figure 2 nanomaterials-14-01295-f002:**
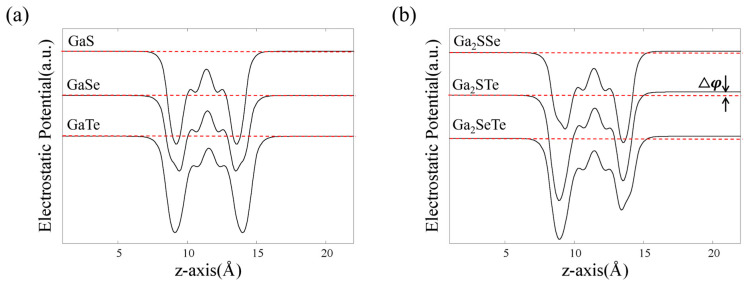
The local potential plots of monolayer (**a**) GaX and (**b**) Janus Ga_2_XY. △φ is the difference between local potentials corresponding to the upper and lower sides of MMCs and Janus MMCs. The red dotted lines represent the vacuum energy of the systems.

**Figure 3 nanomaterials-14-01295-f003:**
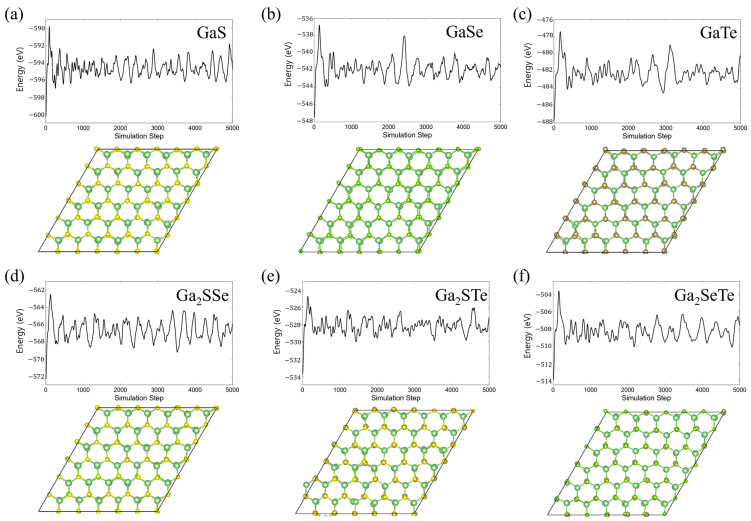
Results of ab initio molecular dynamics (AIMD) calculations. The top and bottom panels represent the total energy and the atomic structure of each system, respectively. The graphs show total energy as a function of time step in units of 1 fs. The atomic structures are the final structures obtained from AIMD calculations.

**Figure 4 nanomaterials-14-01295-f004:**
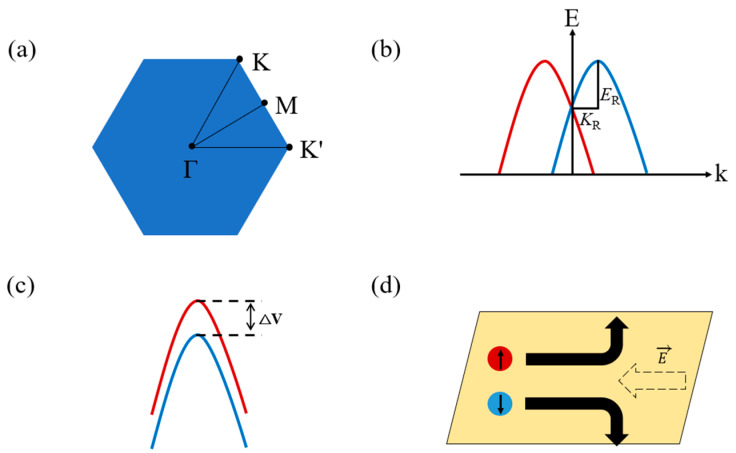
(**a**) The hexagonal Brillouin zone of GaX and Janus Ga_2_XY. Schematic plots of (**b**) Rashba-type spin splitting and (**c**) spin−valley coupling at K or K′ point. (**d**) Schematic plot representing carrier transfer and valley Hall effect. The red and blue lines represent the energy bands of up and down spins, respectively.

**Figure 5 nanomaterials-14-01295-f005:**
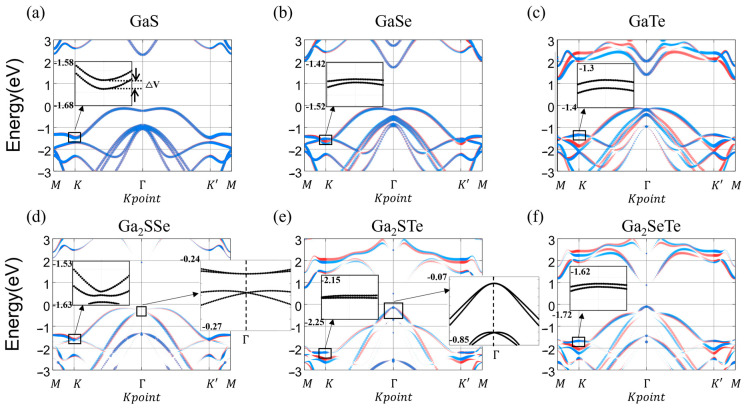
Band structures of monolayer GaX and Janus Ga_2_XY with the SOC effect. The insets represent close-up images around K (or K’) and Γ points. The insets indicate the magnification of the specified regions. The red and blue lines represent the energy bands of up and down spins, respectively.

**Figure 6 nanomaterials-14-01295-f006:**
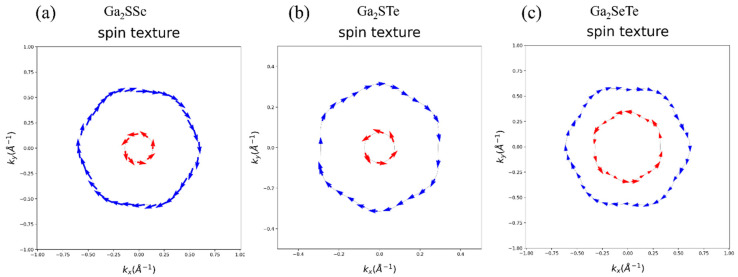
Spin texture of (**a**) Ga_2_SSe, (**b**) Ga_2_STe, and (**c**) Ga_2_SeTe with SOC calculations. From fermi energy, (**a**–**c**) are plotted at positions of −0.4 eV, −0.8 eV, and −1 eV, respectively. The colored arrows represent the spin polarization orientation.

**Figure 7 nanomaterials-14-01295-f007:**
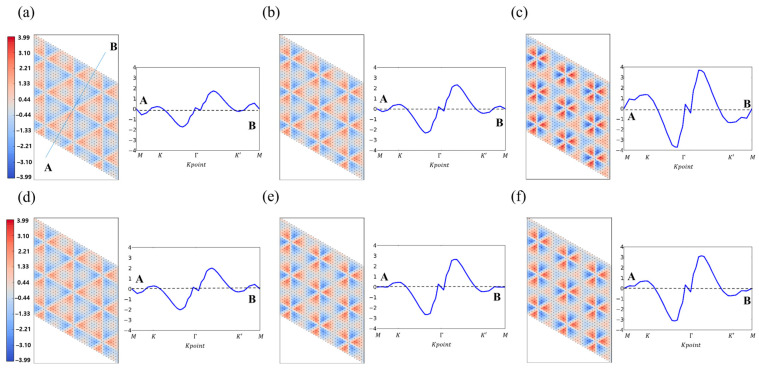
(Left panel) Berry curvatures (Ω(k)) of monolayer GaX and Janus Ga_2_XY. (Right panel) Blue lines indicate the line profiles of Berry curvatures along the AB path.

**Table 1 nanomaterials-14-01295-t001:** The structural parameters and electrical properties of GaX and Janus Ga_2_XY: distances between A and B atoms (*d*_A-B_) and difference in distances between *d*_Ga-X_ and *d*_Ga-Y_ (Δ*d*_Ga-X,Y_), distorted angle (Δ*θ* = *θ*_2_ − *θ*_1_), local potential difference (Δφ), dipole moment (μ), band gap (type), and Zeeman-type spin splitting (ΔV) are given. Here, Δ*θ* is the difference between *θ*_1_ and *θ*_2_, which are the angles denoted in Figure 1b.

	a (Å)	$d_{Ga-Ga}$ (Å)	$d_{Ga-X}$ (Å)	$d_{Ga-Y}$ (Å)	${∆d}_{Ga-X,Y}\;(Å)$	$\theta_{1}$ (deg)	$\theta_{2}$ (deg)	∆θ (deg)	∆φ (eV)	μ (Debye)	Band Gap (eV)	Band Gap Type	∆V (meV)
GaS	3.62	2.45	2.36	2.36	0	117.55	117.55	0	0	–	2.46	Indirect	202
GaSe	3.80	2.45	2.49	2.49	0	118.21	118.21	0	0	–	1.89	Indirect	65
GaTe	4.12	2.45	2.70	2.70	0	118.18	118.18	0	0	–	1.50	Indirect	180
Ga_2_SSe	3.71	2.45	2.39	2.47	0.08	116.07	119.63	3.56	0.25	0.016	2.16	Indirect	97
Ga_2_STe	3.88	2.45	2.44	2.63	0.19	113.27	121.73	8.46	0.61	0.043	0.64	Direct	43
Ga_2_SeTe	3.99	2.45	2.58	2.66	0.08	115.32	120.06	4.74	0.39	0.028	0.98	Direct	65

## Data Availability

Our data that support the findings of this study are available from the corresponding author upon reasonable request.

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
