# Peer review of "Valley-Dependent Electronic Properties of Metal Monochalcogenides GaX and Janus Ga2XY (X, Y = S, Se, and Te)"

_nanomaterials, 2024, doi:10.3390/nano14151295_

Round 1

Reviewer 1 Report

Comments and Suggestions for Authors

In the present work the Authors analyze valley-dependent electronic properties of metal monochalcogenides GaX and Janus Ga2XY (X, Y = S, Se, and Te) materials, at the theoretical level. In particular, they identify the Rashba-type spin splitting in band structures of Janus Ga2SSe and Ga2Ste, as well as confirm that the Zeeman-type spin splitting at the K and K’ valleys of GaX and Janus Ga2XY show opposite spin contributions. The calculate Berry curvatures of GaX and Janus GaXY are also calculated and found that that GaX and Janus Ga2XY have a similar magnitude of Berry curvatures having opposite signs at K and K' points. Moreover, GaTe and Ga2SeTe have relatively larger Berry curvatures than the other GaX and Ga2XY materials, respectively.

The presented paper is well organized and well written. I found no major errors in the presented analysis. The considered subject is timely and presented results may constitute important contribution to the field of 2D materials. There are only several aspects I would like to be additionally addressed before publication: (i) Why should you use the vdw-D3 to study the monolayer systems actually? (ii) I’m missing the structure plot of GaX (iii) How the presented results compare to the leading 2D semiconducting materials, namely the transition metal dichalcogenides, in terms of the valley-properties? There are some qualitative observations that can be made based on the presented results that are not addressed, for example potential of the valley-dependent filtering (please see e.g. Physical Review B 101 (2020), 115423 and Physical Review Letters 124 (2020) 166803 for similar analysis of transition metal dichalcogenide materials)? (v) In general, the text should be improved to provide readers with some comparisons to the valley-physics in the exemplary transition metal dichalcogenides, see good review such as: Nano Research 12 (2019), 2695 or Reports on Progress in Physics 84 (2021) 026401, (iv) There are some grammar, word or spelling errors here and there; the manuscript should be revised one more time in this respect.

Author Response

In the present work the Authors analyze valley-dependent electronic properties of metal monochalcogenides GaX and Janus Ga2XY (X, Y = S, Se, and Te) materials, at the theoretical level. In particular, they identify the Rashba-type spin splitting in band structures of Janus Ga2SSe and Ga2Ste, as well as confirm that the Zeeman-type spin splitting at the K and K’ valleys of GaX and Janus Ga2XY show opposite spin contributions. The calculate Berry curvatures of GaX and Janus Ga2XY are also calculated and found that that GaX and Janus Ga2XY have a similar magnitude of Berry curvatures having opposite signs at K and K' points. Moreover, GaTe and Ga2SeTe have relatively larger Berry curvatures than the other GaX and Ga2XY materials, respectively.

The presented paper is well organized and well written. I found no major errors in the presented analysis. The considered subject is timely and presented results may constitute important contribution to the field of 2D materials. There are only several aspects I would like to be additionally addressed before publication:

Comments 1: Why should you use the vdw-D3 to study the monolayer systems actually?

Response 1: To investigate the GaX and Janus-Ga2XY systems, we employed van der Waals corrected density functional theory (vdW-DFT) with Grimme's D3 dispersion correction. We used the vdW correction because, although they are called monolayer systems, they consist of four atomic layers, and we needed to account for the vdW correction between the atomic layers.

Comments 2: I’m missing the structure plot of GaX.

Response 2: Ga2XY structure consists of Ga atoms bonded with two different chalcogen atoms, whereas GaX comprises Ga atoms bonded with the same type of chalcogen atom. The optimized geometries for GaX were not shown in the manuscript due to its structural similarity to Ga2XY.

Comments 3: How the presented results compare to the leading 2D semiconducting materials, namely the transition metal dichalcogenides, in terms of the valley-properties? There are some qualitative observations that can be made based on the presented results that are not addressed, for example potential of the valley-dependent filtering (please see e.g. Physical Review B 101 (2020), 115423 and Physical Review Letters 124 (2020) 166803 for similar analysis of transition metal dichalcogenide materials)?

Response 3: Based on the presented results, the valley characteristics of GaX and Ga2XY exhibit similarities to those of TMD materials. Specifically, similarities in valley polarization are observed at the K and K' points, including Zeeman-type spin splitting. Moreover, considering the filtering mechanisms discussed in the references the reviewer mentioned, similar mechanisms may potentially exist in the GaX and Ga2XY structures. We incorporate and elaborate on these findings in the paper. We also added the following sentence to the main text:

Main manuscript page 7/12 line 216-224:

… However, the selectively excited states from K and K' by optical valley pumping generate nonequilibrium distributions of carriers, which is related to carrier lifetime due to carrier recombination. Based on recent studies, GaX and Ga2XY monolayers exhibit valley properties comparable to those of TMDs. For instance, the valley-spin filtering mechanism in TMDs is attributed to metal-induced gap states (MIGS). This research contributes to a fundamental understanding and provides design guidelines for efficient valley-spin filter devices based on hexagonal monolayers with broken inversion symmetry [74]. Additionally, the potential of valley-dependent Lorentz forces in strained TMDs to induce valley separation was discussed [75]. This mechanism, coupled with the Zeeman-type spin splitting observed in our GaX and Ga2XY systems, suggests potential applications in valleytronic devices where valley-dependent filtering could be utilized.

The few-layer structure of InSe demonstrates a strong Dresselhaus-type SOC induced by its inversion-asymmetric gamma phase. [76] These features are pivotal for the practical application of SOC effects in Janus materials or alloys. …

Comments 4: In general, the text should be improved to provide readers with some comparisons to the valley-physics in the exemplary transition metal dichalcogenides, see good review such as: Nano Research 12 (2019), 2695 or Reports on Progress in Physics 84 (2021) 026401

Response 4: We really appreciate the reviewer’s comment. In line with the comment, we added the sentence in main text, as follows:

Main manuscript page 2/12 lines 64 - 66:

… Based on various methods of controlling electrons through valleys including the two methods mentioned above, 2D materials like TMDs and janus TMDs (MXY, X, Y=calcogen) have been proposed as candidates for valleytronics. [42-51] The relation between valleytronics and TMD materials was thoroughly explored in the previous studies [52, 53]. However, until now most studies of valleytronics of 2D materials have been limited to TMDs, while there are few reports of other 2D materials like metal monochalcogenides (MMCs). …

Comments 5: There are some grammar, word or spelling errors here and there; the manuscript should be revised one more time in this respect.

Response 5: We carefully revised to correct any grammatical, spelling, and word usage errors.

Reviewer 2 Report

Comments and Suggestions for Authors

The study by Kim J.et al. is in interesting and carefully executed study, which explores the valley degree of freedom of GaX and Janus GaXY (X, Y = S, Se, 12
Te). The used techniques and metholodogical approaches are proper and well executed, the topic has plenty of novelty and it also fits the scope of nanomaterials therefore I recommend the publication of the work, after a minor revision. I don't have questions, but please find my comments below and I suggest that you try to work some of it into the revised manuscript.

1. The description of the Theoretical method and formulas used  is too short and can be described in greater detail.

2. The conclusions are well sustained by the experimental data shown and of interest to those studying

Comments on the Quality of English Language

Minor editing of English language required

Author Response

The study by Kim J.et al. is in interesting and carefully executed study, which explores the valley degree of freedom of GaX and Janus Ga2XY (X, Y = S, Se, and Te). The used techniques and metholodogical approaches are proper and well executed, the topic has plenty of novelty and it also fits the scope of nanomaterials therefore I recommend the publication of the work, after a minor revision. I don't have questions, but please find my comments below and I suggest that you try to work some of it into the revised manuscript.

Comments 1: The description of the Theoretical method and formulas used is too short and can be described in greater detail.

Response 1: Thank you for the reviewer’s kind comments. We improved the manuscript by providing more comprehensive explanations of the theoretical methodologies and formulas employed. The “Calculation Methods” were revised as follows:

Main manuscript page 2/12 line 92 – page 3/12 line 104:

For investigations of GaX and janus Ga2XY, we have performed DFT calculations within the generalized gradient approximation (GGA) for the exchange-correlation (xc) functionals [54,55], implemented in the Vienna ab initio simulation package (VASP) [56,57]. The kinetic energy cut-off is set to 400 eV and the projector-augmented wave potentials (PAW) are used to treat the ion-electron interaction [58,59]. For the van der Waals (vdW) corrections, we use Grimme's DFT-D3 method based on a semiempirical GGA-type theory [60]. To remove artificial interaction between the adjacent slabs, we use a vacuum region of about 20Å. The SOC effect is considered in electronic structure calculations. In the DFT calculations, we consider (1 × 1) hexagonal unit cells of monolayer GaX and Janus Ga2XY. Furthermore, we use the (12 × 12 × 1) grid in the gamma-centered scheme for the Brillouin zone integration. The Hellmann–Feynman forces and energy convergence criteria are set to 0.01 eV/AÌŠ and 10-5 eV for ionic relaxation, respectively. Moreover, to study the thermal stability of GaX and Janus Ga2XY, we perform ab initio molecular dynamics (AIMD) simulation. The NVT ensemble is considered, and the simulation temperature is set to 300K. The (6 x 6) hexagonal unit cell for AIMD simulation is used, and total simulation time is set to 5000 fs and each time step is 1fs.

The Berry curvature is a fundamental physical quantity that plays a pivotal role in the study of topological materials. The Berry curvature  Ωij is defined as Ωij = iAj - ∂jAi, where i ​and​ j are partial derivative operators with respect to parameters in the parameter space, and Ai and Aj​ are the components of the Berry connection. The Berry connection Ai is defined as Ai =i<u(k)|iu(k)>, where |u(k)> represents the periodic part of the Bloch state, which encapsulates the geometric properties of the electronic state, and k is the wavevector in the Brillouin Zone. The Berry curvature can be expressed using Bloch states as follows [61-63]:

Ωij = i(<iu(k)|ju(k)> - <ju(k)|iu(k)>),

In our calculations, we use VASPBERRY software [64] to investigate the Berry curvature for the Janus GaX and Ga2XY materials.

Comments 2: The conclusions are well sustained by the experimental data shown and of interest to those studying.

Response 2: I really appreciate the reviewer’s valuable support and encouragement.